# Evaluation of peripheral intravenous line securement devices under clinically relevant loading and perspiratory conditions

**George K. H. Morgan**[ID]**, Maegan Spiteri, Spyros Masouros**[ID]*

Department of Bioengineering, Imperial College London, London, United Kingdom

* s.masouros04@imperial.ac.uk

## Abstract

Peripheral vascular access devices (PVADs) and their associated intravenous (IV) lines are widely used in clinical care. Securement devices are typically employed to prevent accidental dislodgement and complications such as occlusion and phlebitis. Despite the wide use of securement devices, there are no standardised methods for evaluating their actual ability to secure IV lines. This study assessed the securing ability of four commercially available IV line securement devices – Grip Lok, Javelo, micropore tape, and IV bracelet – under controlled laboratory conditions. First, all four devices were tested for resistance to axial pull-out forces using a uniaxial materials testing machine across ten participants. Participant comfort during testing was assessed via questionnaire. Grip-Lok and Javelo exhibited significantly greater axial pull-out strength than the other two devices (p-adj < 0.001), and were progressed to further testing, consisting of off-axis loading at 90°, 135°, and peel-off, and axial pull-out under simulated perspiration conditions. No significant differences were observed between Grip Lok and Javelo under loading in 90° and 135°. Javelo demonstrated superior performance in both peel-off and perspirant conditions (p-adj < 0.05). Javelo was rated the most comfortable by 60% of participants. The results highlight the need for incorporating realistic loading conditions and comfort metrics when evaluating the performance of infusion line securement aids. Furthermore, the results provide evidence that using a non-adhesive based device, particularly in patients with marked diaphoresis might provide more infusion line security, though further study is required.

## 1. Introduction

Peripheral vascular access devices (PVADs), such as peripheral intravenous catheters (PIVCs) and arterial lines, enable clinicians to deliver medical treatments and interventions rapidly. PVADs are the most frequently performed invasive medical procedure; an estimated 350 million PIVCs alone are purchased in the United States annually [1]. It is estimated that 60–90% of hospitalised patients require a PIVC

**Data availability statement:** All force and displacement data from the Stage 1 and Stage 2 testing, as well as the results of the comfort survey, are available at Zenodo: https://doi.org/10.5281/zenodo.16317677.

**Funding:** The consumables, arm rest, and devices used in this study were provided courtesy of Javelo Health. GM was supported by a DTP studentship from the Engineering and Physical Sciences Research Council (EP/T51780X/1). The mechanical testing was conducted in the EPSRC Injury and Reconstruction Biomechanics Test Suite (EP/S021752/1).

**Competing interests:** The authors have declared that no competing interests exist.

during their care [2]. However, the effective securement of PVADs to patients remains a clinical challenge. PVAD failure rates of approximately 38–43% [3] are reported, with the most prominent failure types being occlusion or infiltration, dislodgement, and phlebitis [4–7]. These complications can delay treatment administration, increase infection risk and patient discomfort, and increase provider costs [8,9].

PVAD failure is influenced by the stability of connected intravenous (IV) treatment lines. Forces on these lines, for example due to an accidental pull or tangling, can destabilise the PVAD leading to complications or dislodgement [5–7,10–12]. Accordingly, securement devices can be used to anchor IV lines to the patient thus reducing the probability of PVAD failure and its consequences [7]. These securement devices are generally adhesive-based [3,13]. Various standards and guidelines recommend the use of such securement devices to prevent PVAD failure [13,14]. The need to enhance the performance of these securement devices has also been recognised [3,4], underscoring the importance of comparative testing methods to assess whether newer devices offer improvements over existing devices.

Despite the clinical importance of securement devices, standardised mechanical testing methodologies to evaluate their performance in securing IV lines remain underdeveloped. Studies investigating the mechanical performance of securement devices are limited, and often test only axial pull-out [15,16] or time-based catheter survival rates in clinical settings [17]. Additionally, clinically unwell patients are often perspirant, which healthcare staff have identified as a common factor leading to dislodgement [18]; yet, the effect of perspiration on maintaining adequate levels of securement has not been considered in tests. Furthermore, the success of a securement device in a clinical setting depends on patient acceptance and uptake, which is greatly impacted by the relative comfort of devices, yet patient comfort data for securement devices has not been reported.

This study first compared four commercially-available securement devices in terms of their mechanical resistance to axial pull-out forces, as previous studies have done. Two of the devices were subsequently tested for their mechanical resistance to forces applied at various angles, and for their resistance to axial pull-out forces in simulated perspiratory conditions. Participant comfort during axial pull-out testing was assessed using a questionnaire.

## 2. Methods

### 2.1 Experimental protocol

Ethical approval for this study was obtained from the Imperial College Science Engineering Technology Research Ethics Committee (number: 6590569). Ten healthy adult participants were recruited for the study between 6th November 2024 and 14th January 2025 and written informed consent was granted prior to testing. Exclusion criteria included: history of musculoskeletal injury in the hand or wrist area, known allergy to adhesives, and skin or connective tissue disorders.

**2.1.1 Stage 1.** Four commercially available IV line securement devices were tested: micropore tape (3M, St. Paul, United States), Grip-Lok (TIDI Products, Neenah, United States), IV bracelet (Medow, Lund, Sweden), and Javelo Line

Securement Gen 1 (Javelo Health, Edinburgh, United Kingdom). Each of the devices is shown applied according to manufacturer instructions in Fig 1.

IV line segments (Intrafix Primeline; B. Braun, Melsungen, Germany) 85 mm in length and 4.1 mm in outer diameter were attached to a uniaxial materials testing machine (5866; Instron, Canton, USA), passed through a bespoke pulley fixture, and secured to participants' right arms using IV line securement devices. The participants were then asked to rest their arms on an adjustable-height arm rest (Phlebotomy Arm Rest; Meditelle, Birmingham, United Kingdom). The arm rest was adjusted to align the IV line at 0°, such that the IV line lay straight through the device. For the Javelo device, this meant the arm was perpendicular to the IV line, as shown in Fig 1d. The test set-up is shown in Fig 2.

The IV line was pulled at a rate of 8 mm s$^{-1}$ to a displacement of 150 mm, or until a maximum force of 20 N was reached. Force and displacement were recorded at the testing machine's load cell. Each of the four devices was tested on the 10 participants, for a total of 40 tests. Device performance in Stage 1 was used to select devices to be tested in Stage 2 to reduce experiment time.

**2.1.2 Stage 2.** Devices which performed favourably in Stage 1 trials were subsequently taken forward in Stage 2, whereby the devices were tested on the right arm of each participant with the IV line at an angle to the device, and in pull-out under induced perspiration.

Testing at an angle: A similar testing setup as shown in Fig 2 in Stage 1 trials was used, but the arm rest was positioned such that the IV line passed through the devices at an angle of 90° or 135°, as shown in Fig 3a and 3b, respectively, or the arm rest was aligned in the pull-out position, but with the pulley fixture removed, simulating a 'peel-off' scenario, as shown in Fig 3c. These loading conditions were selected to capture a range of possible realistic scenarios during clinical use.

Perspiration: One of each device was applied to each arm of each participant, with the devices randomly assigned to the arms, and the participants were asked to perform cardiovascular exercises of their choice to induce perspiration. Perspiration was detected using water contact indicator tape (3M, St. Paul, United States) which was applied to the arms of each participant next to the devices. Once perspiration was detected as shown in Fig 4, the participants were asked

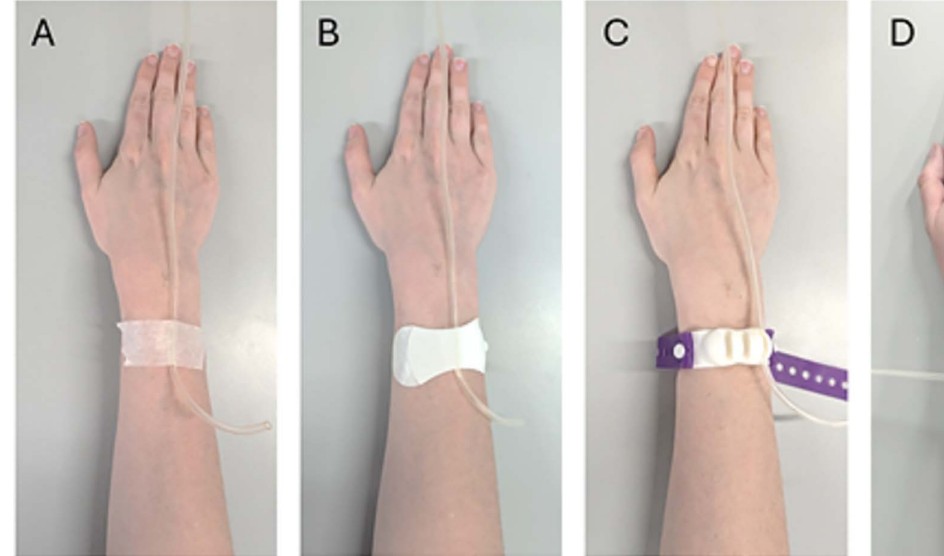

**Fig 1. Securement devices.** The four IV line securement devices tested, shown applied to the arm according to manufacturer instructions and with IV line secured in the device in a linear line securement arrangement: **(a)** micropore tape, **(b)** Grip-Lok, **(c)** IV bracelet, **(d)** Javelo.

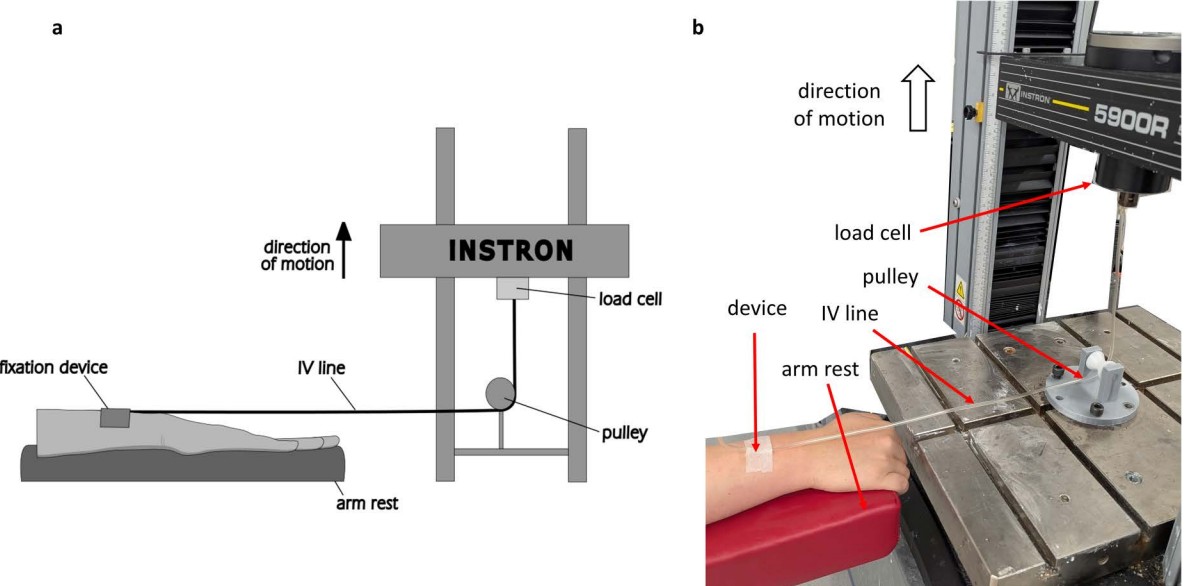

**Fig 2. Pull-out testing setup. (a)** Diagram of the test setup for a pull-out scenario in a uniaxial materials testing machine. Load cell, bespoke pulley fixture, device, IV line, and arm shown. **(b)** Photograph of the pull-out testing setup with micropore tape as the device used.

to continue to exercise for five additional minutes, and then the pull-out test from Stage 1 was performed. For all Stage 2 tests, IV lines were pulled at a rate of 8 mm s$^{-1}$ to a displacement of 150 mm, or until a maximum force of 20 N was reached. Each of the devices was tested on 10 participants.

## 2.2 Comfort survey

Participants were asked the following to assess their comfort during the Stage 1 trials, and their responses were recorded: "Please provide a ranking for each of the tested devices based on your perception of their comfort during the testing, with 1 being the most comfortable and 4 being the least comfortable".

## 2.3 Data analysis

**2.3.1 Stage 1.** A Friedman test was performed on the mean peak force recorded for each of the four devices. Post-hoc Wilcoxon signed-rank tests with a Benjamini-Hochberg multiple-comparisons correction procedure were performed to compare the mean peak forces for each pair of devices. A significance threshold of $\alpha < 0.05$ was used.

**2.3.2 Stage 2.** One-tailed Wilcoxon signed-rank tests with a Benjamini-Hochberg multiple-comparisons correction procedure were performed to compare the mean peak force recorded for the devices included in Stage 2 trials for each of the four tests performed: 90°, 135°, peel-off, and perspirant pull-out. A significance threshold of $\alpha < 0.05$ was used.

## 3. Results

The peak forces for each device during the Stage 1 trials and the results of a Friedman test with post-hoc Wilcoxon signed-rank tests between each device pair are shown in Fig 5. There were significant (p-adj < 0.001; p-value adjusted using Benjamini-Hochberg correction procedure for multiple comparisons) differences in the mean peak forces for every device pair except between the Grip-Lok and Javelo, which had no significant difference in mean peak forces (p-adj > 0.05; p-value adjusted using Benjamini-Hochberg correction procedure for multiple comparisons). The Grip-Lok and Javelo had the highest mean peak forces in Stage 1 trials, so these devices were progressed to Stage 2 trials.

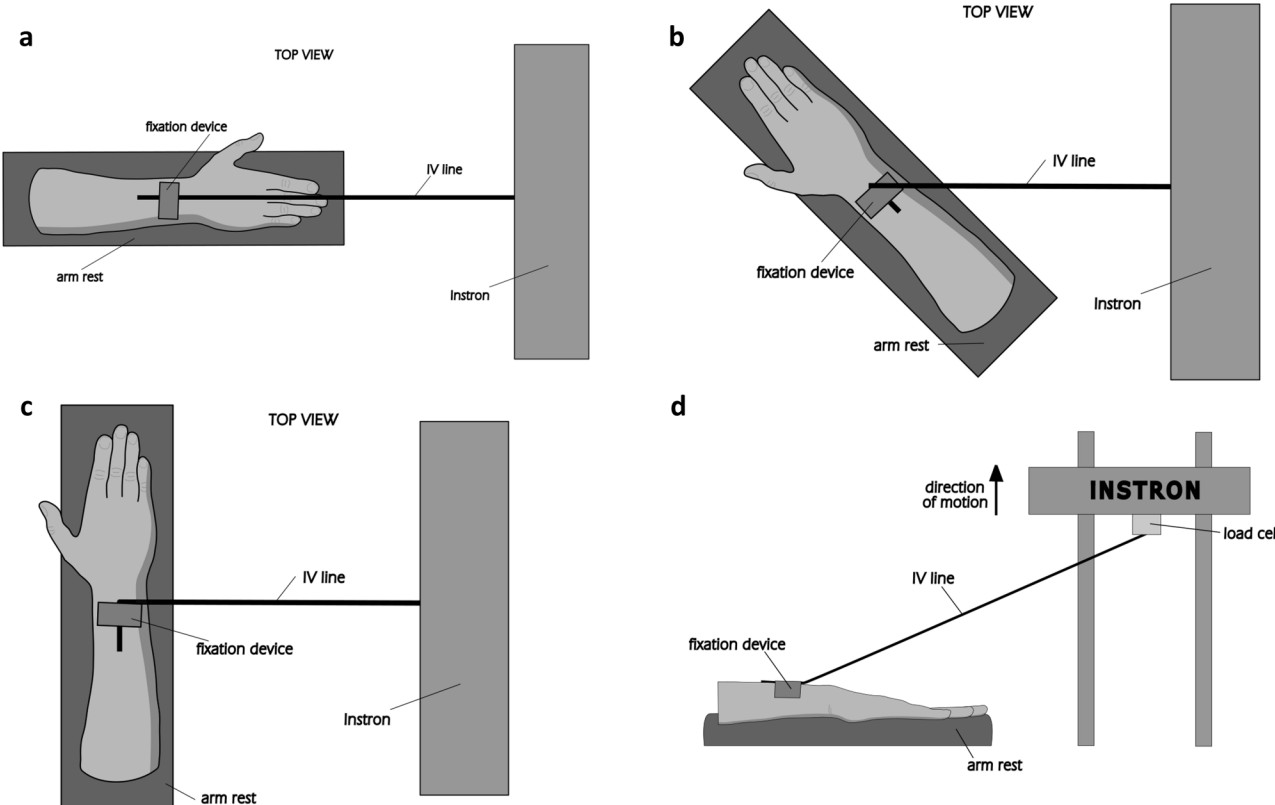

**Fig 3. Diagrams of the different testing setups. (a)** Bird's eye view of the pull-out test, **(b)** bird's eye view of the 135° test, **(c)** bird's eye view of the 90° test, **(d)** peel-off testing setup, which is identical to the pull-out test but with the pulley fixture removed.

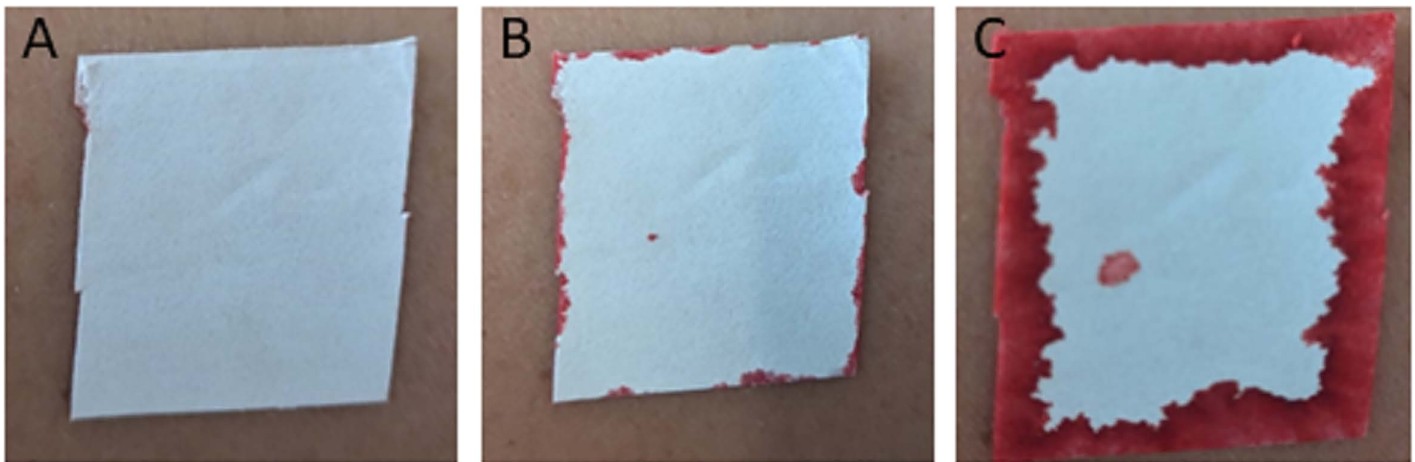

**Fig 4. Perspiration indicator.** Water contact indicator tape applied to participant's arm next to device. Indication of perspiration was identified by a solid red line around the full-perimeter of the tape segment, with red penetrating towards the centre of the tape segment: **(a)** before cardiovascular exercise, **(b)** part-way through exercise with partial indication, **(c)** full indication at which point participants performed five additional minutes of exercise before testing commenced.

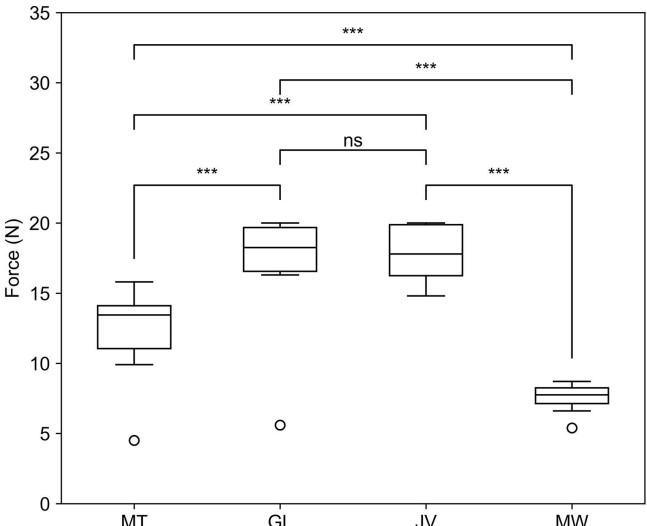

**Fig 5. Stage 1 results.** Box and whisker plots of peak forces during pull-out (Stage 1) testing for each of the four devices tested: micropore tape, Grip-Lok, Javelo, and Medow. The results of a Friedman test with post-hoc Wilcoxon signed-rank tests with a Benjamini-Hochberg correction procedure are displayed. n = 10. ***p-adj < 0.001; ns = no significance.

The peak forces for the Grip-Lok and Javelo in the four testing methods in Stage 2 trials, along with the results of a one-tailed Wilcoxon signed-rank test between the Grip-Lok and Javelo devices for each of the four testing methods are shown in Fig 6. The mean peak force of the Javelo was significantly greater (p-adj < 0.05; p-value adjusted using

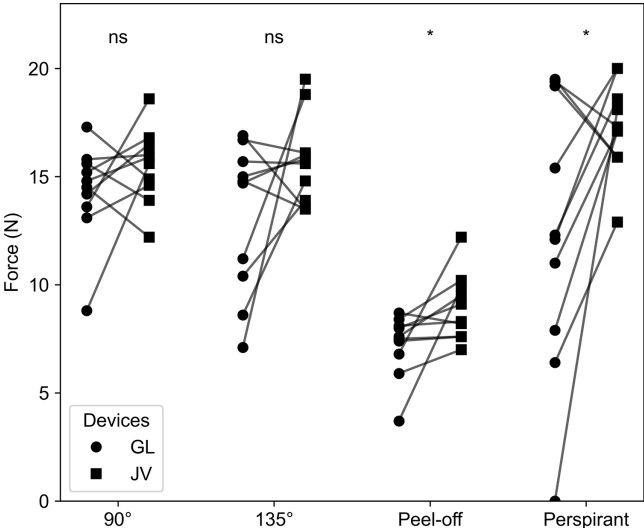

**Fig 6. Stage 2 results.** Paired plots of peak forces during the angled and perspiration pull-out (Stage 2) testing: 90°, 135°, peel-off, and perspiration pull-out. The Grip-Lok and Javelo devices were tested. The results of one-tailed Wilcoxon signed-rank tests with a Benjamini-Hochberg correction procedure between the devices for each of the four tests are displayed. n = 10. *p-adj < 0.05; ns = no significance.

Table 1. Comfort survey results. The results of the comfort survey completed by participants after Stage 1 trials. The average rank for each device (where 1 is most comfortable and 4 is least comfortable) and the proportion of participants that ranked each device as 'most comfortable' are shown. n = 10.

| Device | Micropore tape | Grip-Lok | Medow | Javelo |
|---|---|---|---|---|
| Average rank | 3.3 | 2.8 | 2.2 | 1.7 |
| Ranked most comfortable | 0% | 20% | 20% | 60% |

Benjamini-Hochberg correction procedure for multiple comparisons) than that of the Grip-Lok for the peel-off and perspirant pull-out tests. There was no significant difference between the devices in the 90° and 135° angled pull-out tests.

The results of the comfort survey, which was completed by participants immediately after Stage 1 trials, are shown in Table 1. The average rank of each device across the 10 participants, as well as the proportion of participants that ranked each device as the most comfortable, are shown.

The data resulting from the Stage 1 trials, Stage 2 trials, and comfort survey are publicly available in a Zenodo depository [19].

## 4. Discussion

The results of this study demonstrate that the Grip-Lok and the Javelo were the superior options for IV line securement in axial pull-out testing, with both devices exhibiting significantly greater resistance to dislodgement compared to their adhesive-based counterparts, micropore tape and the Medow IV bracelet. This finding aligns with previous studies highlighting the importance of securement devices in reducing PVAD failure rates and underscores the necessity of robust mechanical performance in clinical practice [3,4].

When evaluating securement performance under angled loading conditions, no significant difference was observed between the Grip-Lok and the Javelo in the 90° or 135° orientations. This suggests that both devices provide comparable resistance to dislodgement when subjected to off-axis forces, which is a relevant consideration in clinical settings where IV lines are frequently exposed to multidirectional forces due to patient movement. While axial pull-out testing is the accepted method for securement devices, this test does not fully encapsulate the range of mechanical environments experienced in clinical settings. Axial pull-out may be used as a test to exclude underperforming devices, but resistance to axial forces alone is not sufficient to determine overall device efficacy. While all devices investigated in this study were tested with a linear line securement arrangement, as shown in Fig 1, future studies should investigate the performance of devices with 'looped' IV lines, as is sometimes recommended for clinical use and may affect pull-out resistance performance.

The Javelo device demonstrated superior performance to the Grip-Lok in both peel-off and perspirant conditions, with significantly greater peak force required for detachment. Given the widespread use of adhesive-based securement devices, these findings emphasize the critical influence of skin moisture on device performance. The reduction in securement strength observed with the Grip-Lok under perspiration highlights the potential for device failure in febrile or diaphoretic patients, where adhesive degradation may compromise line stability. These findings suggest that non-adhesive devices should be recommended instead of adhesive-based devices for use on potentially febrile or diaphoretic patients, such as those with infection, acute medical illness, or traumatic injury, though further investigation is needed. Additionally, as perspirant conditions can significantly affect device performance, and are likely to be encountered clinically, mechanical testing aimed to qualify securement devices should include testing under perspirant conditions.

An additional, qualitative observation from this study was that the Grip-Lok exhibited reduced effectiveness in participants with substantial arm hair, whereas the Javelo appeared unaffected. Although arm-hair density was not recorded, the qualitative observation suggests that hair density may be a confounding factor in securement performance, particularly for adhesive-based devices. Future studies should consider including hair-density measurements to assess its impact

on mechanical securement, thereby informing design adaptations for more effective universal application across diverse patient populations. Furthermore, differences in device performance when applied to the left or right arm, or to the dominant or non-dominant arm, were not measured in this study. Arm dominance may affect the performance of these devices due to differences in size, perspiration, and hair density. Future studies should investigate the effect of arm dominance on device performance, as devices are most commonly applied to the non-dominant arm in clinical settings.

Beyond mechanical performance, comfort is a critical determinant of patient adherence and device acceptability. The subjective rankings collected in this study indicated that the Javelo was the most preferred device, with 60% of participants ranking it as the most comfortable option. The Javelo and the Medow IV bracelet were rated as more comfortable than the micropore tape and the Grip-Lok, suggesting that adhesive devices are less comfortable than non-adhesive devices. Future device development and testing should consider and report patient comfort as a key metric, as improved comfortability will increase clinical compliance and long-term catheter retention.

This study addresses the mechanical performance and testing of securement devices, but other mechanisms of PVAD failure, such as phlebitis and occlusion, remain unaddressed. A large-scale clinical trial comparing all-cause PVAD failure rates when using different securement devices should be performed. Data from sub-populations, such as those likely to perspire during treatment, can additionally address specific securement device recommendations for these sub-populations. This study is a step towards addressing the overall problem of PVAD failure and can be used to inform device selection in a future clinical trial, as well as to inform methodologies of future mechanical studies investigating PVAD dislodgement, and regulatory mechanical testing protocols.

## 5. Conclusions

This study is the first to compare IV line stabilisation devices under a range of loading conditions expected in clinical scenarios, including angled force application and testing under perspirant conditions. The devices tested experienced a large range of axial pull-out force. Two of the four devices, one adhesive-based and one non-adhesive based, were able to withstand expected forces without dislodgement under all loading conditions, but the adhesive-based one was not able to withstand forces under perspiration conditions. These findings should be investigated further with a larger sample size, and further parameters should be investigated, such as the effect of the line securement arrangement used and the influence of hand-dominance on device stability. The findings of this study highlight the importance of testing securement devices beyond standard axial pull-out methodologies. As perspiration was found to impact significantly adhesive-based securement, future mechanical assessments should consider incorporating moisture conditions to ensure robust device performance in diverse patient populations and clinical settings.

## Acknowledgments

The mechanical testing was conducted in the EPSRC Injury and Reconstruction Biomechanics Test Suite. For the purpose of open access, the authors have applied a Creative Commons Attribution (CC BY) license to any Author Accepted Manuscript version arising.

## Author contributions

**Conceptualization:** George K. H. Morgan, Spyros Masouros.

**Data curation:** George K. H. Morgan, Maegan Spiteri.

**Formal analysis:** George K. H. Morgan.

**Investigation:** George K. H. Morgan, Maegan Spiteri.

**Methodology:** George K. H. Morgan, Spyros Masouros.

**Project administration:** Spyros Masouros.

**Supervision:** Spyros Masouros.

**Visualization:** George K. H. Morgan, Maegan Spiteri.

**Writing – original draft:** George K. H. Morgan, Spyros Masouros.

**Writing – review & editing:** George K. H. Morgan, Maegan Spiteri, Spyros Masouros.

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
