## [Decision Letter · Decision Letter 0]

16 Nov 2025

Dear Dr. Masouros,

Thank you for submitting your manuscript to PLOS ONE. After careful consideration, we feel that it has merit but does not fully meet PLOS ONE’s publication criteria as it currently stands. Therefore, we invite you to submit a revised version of the manuscript that addresses the points raised during the review process.

We look forward to receiving your revised manuscript.

Kind regards,

Federica Canzan

Academic Editor

PLOS ONE

Journal Requirements:

The consumables, arm rest, and devices used in this study were provided courtesy of Javelo Health. GM was supported by a DTP studentship from the Engineering and Physical Sciences Research Council (EP/T51780X/1). The mechanical testing was conducted in the EPSRC Injury and Reconstruction Biomechanics Test Suite (EP/S021752/1). For the purpose of open access, the authors have applied a Creative Commons Attribution (CC BY) license to any Author Accepted Manuscript version arising.

The consumables, arm rest, and devices used in this study were provided courtesy of Javelo Health. GM was supported by a DTP studentship from the Engineering and Physical Sciences Research Council (EP/T51780X/1). The mechanical testing was conducted in the EPSRC Injury and Reconstruction Biomechanics Test Suite (EP/S021752/1).

4. Please amend your authorship list in your manuscript file to include author Spyros Masouros.

5. Please amend the manuscript submission data (via Edit Submission) to include author Spyros D. Masouros.

6. Please amend your list of authors on the manuscript to ensure that each author is linked to an affiliation. Authors’ affiliations should reflect the institution where the work was done (if authors moved subsequently, you can also list the new affiliation stating “current affiliation:….” as necessary).

7. We note that Figure(s) 1, 2, 3, 4 in your submission contain copyrighted images. All PLOS content is published under the Creative Commons Attribution License (CC BY 4.0), which means that the manuscript, images, and Supporting Information files will be freely available online, and any third party is permitted to access, download, copy, distribute, and use these materials in any way, even commercially, with proper attribution. For more information, see our copyright guidelines: http://journals.plos.org/plosone/s/licenses-and-copyright.

a. You may seek permission from the original copyright holder of Figure(s) 1, 2, 3, 4 to publish the content specifically under the CC BY 4.0 license.

Reviewers' comments:

Reviewer's Responses to Questions

**Comments to the Author**

1. Is the manuscript technically sound, and do the data support the conclusions?

Reviewer #1: Partly

Reviewer #2: Partly

2. Has the statistical analysis been performed appropriately and rigorously?

Reviewer #1: I Don't Know

Reviewer #2: No

3. Have the authors made all data underlying the findings in their manuscript fully available?

Reviewer #1: Yes

Reviewer #2: No

4. Is the manuscript presented in an intelligible fashion and written in standard English?

Reviewer #1: Yes

Reviewer #2: Yes

Reviewer #1: General comment:

Effective stabilisation and securement of intravenous peripheral vascular access devices (PVAD - the cannula, insertion needle, integrated components – wings, needle-free connectors etc.) can help to reduce therapy failure and have become important topics of research and feature in preventative care bundles.

Infusion lines when connected to PVADs and infusion line ‘addons’ (in-line filters, stopcocks etc.) are implicated in complications like mechanical phlebitis and accidental dislodgment, particularly amongst mobile, physically active or confused patients. However, unless they are part of a closed infusion system, they are not generally considered as the PVAD itself.

Confusingly the article refers to PVAD stabilisation/securement and the securement of the associated infusion lines as the same. Whilst intimately related the two are not the same and should be differentiated. Whilst I accept that this situation might reflect the relative sparsity of research literature on the securement of infusion lines (as acknowledged in the article - many of the supporting citations refer to PVAD stabilisation or securement) it should be clarified by the authors that the study is about infusion line securement and not PVAD securement.

Specific comments:

Title:

Rather than referring to’ access line’ in the title it might be better phrased as ‘infusion line’ to clarify you are not referring to needle-free connectors or other.

Abstract:

Lines 35-37 I am not sure if the conclusions can be stated as unqualified as the text suggests. It might be better phrased more tentatively as this example: ‘The results highlight the need for incorporating realistic loading conditions and comfort metrics when evaluating the performance of infusion line securement aids. Furthermore, the results provide evidence that using a non-adhesive based device, particularly in patients with marked diaphoresis might provide more infusion line security, though further study is required.

Introduction:

Line 52-54, needs supporting citations (lots available) which will support the importance of your research.

Lines 56- 57, citation number 7 is incorrect in the reference list. Angie Malone was one of the reviewers for the Infusion Nurses Society standards publication. The correct citation is probably: Nickel B, Gorski L, Kleidon T, Kyes A, DeVries M, Keogh S, et al., Infusion therapy standards of practice. 9th ed. J Inf Nurs, 2024. 47(1S): p. S1-285. doi:10.1097/NAN.0000000000000532. Please check and clarify.

citation number 8 is concerned with central vascular access and not peripheral vascular access – does it also consider the securement of infusion lines? Clarify.

Methods:

Line 80, presumably healthy adult patients (they were asked to undergo physical activity to create sweating – clarify.

Line 84 stage 1 and line 109, of the four devices applied to all 10 participants were they distributed between both left and right arms or was one used in preference? Clarification might support decision in line 127.

Lines 85-87, it might be required to recognise trademarks™® and copyright© in the text, check journal author guidelines. 3M products are now marketed under the company name Solventum in the UK.

Line 88, Figure 1 suggests that all the devices were testing with a linear line securement arrangement – this is fine for the experimental set up, but some important confounding variables are introduced with this approach. For example, many using adhesive based aids in practice loop the lines to ensure better and more comfortable line security. In addition, the IV bracelet instructions for clinical use suggest looping the line back though the holder. This might impact on pull resistance. Please expand on this in a consideration of study limitations in your discussion.

Line 94, State the outer diameter of the experimental infusion line to confirm that it is compatible with the design parameters of the tested devices.

Line 113, why was this done? Time practicalities of the experimental set up, availability of participants or do you have any evidence (or theory) that devices that perform less will under uniaxial forces also perform less well under angular (135, 90 degree forces)? Comment.

Line 127, did stage 1 results support a view that there were no differences between either arm (hairiness reported to affect tape adhesion -line 226 – arms might not be equally hirsute). Comment.

Results:

Line 189, more information about the ‘questionnaire’/ ‘comfort survey’ would be welcome. For example, how sophisticated was this tool? How were participants asked about comfort to enable them to rank it? Comfort during application, wearing, removal, visual appearance etc.

Discussion:

Please add a more detailed consideration of ‘Study limitations and recommendations for future work’ to the discussion.

Conclusion:

Please see comments re abstract.

References:

Require revision to presented in the journal style.

Infusion line securement when done effectively can ensure infusion system integrity, prevent entanglements and reduce PVAD and infusion related complications. It is an understudied and underappreciated aspect of safe infusion therapy and articles like this are to be welcomed.

Whilst many clinical settings will have preferred products and arrangements for peripheral infusion scenarios studies like this one can add to the development of meaningful evidence to enable clinicians to compare experimental outcomes with (future) real world performance of their approaches.

Reviewer #2: The research titled “Evaluation of Peripheral Vascular Access Line Securement Devices Under Clinically Relevant Loading and Perspiratory Conditions” addresses a highly relevant and important topic that warrants further investigation.

However, the study seems small-scale experiments rather than a comprehensive research effort. In the introduction, the authors note that “the effective securement of PVADs to patients remains a clinical challenge,” citing PVAD failure rates of approximately 38–43%, primarily due to occlusion, infiltration, dislodgement, and phlebitis. Despite this, the study focuses primarily on resistance to load and patient comfort, which only partially reflect the broader clinical challenges identified. Consequently, the problem statement, research objectives, and results appear misaligned.

Furthermore, the sample size, including both the number of participants and the types of tests performed, seems insufficient to draw meaningful conclusions about the overall effectiveness of PVAD securement devices.

In the results section, the authors report that “the mean peak force of the Javelo was significantly greater (p-adj < 0.05; p-value adjusted using the Benjamini–Hochberg correction procedure for multiple comparisons) than that of the Grip-Lok for the peel-off and perspirant pull-out tests. There was no significant difference between the devices in the 90° and 135° angled pull-out tests.” However, these reported significant differences were not clearly reflected in Figure 6

Overall, while the research topic is both interesting and potentially impactful, the study’s methodology, experimental design, and data interpretation appear weak and require substantial improvement to support the stated conclusions.

**Do you want your identity to be public for this peer review?** For information about this choice, including consent withdrawal, please see our Privacy Policy

Reviewer #1: No

Reviewer #2: No

---

## [Author Response · Author response to Decision Letter 1]

12 Dec 2025

Response to editor comments:

We thank the editor for their comments. Please find below a response to each of the points raised.

1. The manuscript has now been edited to match PLOS One’s style requirements, including those of file naming.

2. The data repository is already uploaded and is available to reviewers and editors at the Data Review URL provided in the file system. Reference to this repository has been added to the manuscript text and references. The repository is currently restricted such that only those with this Data Review URL (editors and reviewers) are able to access it. On acceptance of the manuscript, the repository will be changed from 'restricted' to 'open' access. This change will take effect instantly.

3. All funding has been removed from the acknowledgements section of the manuscript. The current funding statement is correct.

4. Author ‘Spyros Masouros’ is listed on the manuscript as author ‘Spyros D. Masouros’.

5. Author ‘Spyros Masouros’ has been amended to ‘Spyros D. Masouros’ in the manuscript submission data.

6. Affiliation information has now been added for all authors in the manuscript.

7. No copyright images are contained in any of the figures in this submission. Figures 1 and 4 consist of photos taken by and of the authors during preliminary testing. Figures 2 and 3 were also created by the authors using photos taken by and of the authors during preliminary testing, and diagrams (all part of the diagram, including the hand, rest, and testing machine) which were drawn by the authors.

8. This has been noted. No specific works were recommended by the reviewers. Several references have been added to the manuscript.

Response to reviewer 1:

We thank the reviewer for their comments. We hope that the changes in the manuscript and the responses below sufficiently address the reviewer’s concerns.

Title:

The title has been changed to ‘infusion line’ rather than ‘access line’.

Abstract:

Lines 35-37: the text has been changed as recommended.

Introduction:

The distinction between PVAD and IV line and securement has now been clarified throughout the manuscript, and it has been made more clear that this study investigates IV line securement.

Lines 52-57: Additional citations have been added, and citation number 8 has been removed as it does not consider the securement of peripheral vascular access.

Methods:

Line 80: The text has been amended to specify that the participants were healthy adults.

Lines 85-87: The PLOS One author guidance states that PLOS One “cannot publish copyright symbols such as ©, ®, or ™.” The water contact indicator tape used was purchased from ‘3M’ and no equivalent product is available under Solventum, so we have left the manufacturer name as ‘3M’ to aid in reproducibility.

Line 88: The following text has been added to the discussion section to state the limitation of this study to linear line securement arrangements, and to suggest that future studies investigate alternative arrangements:

“While all devices investigated in this study were tested with a linear line securement arrangement, as shown in Figure 1, future studies should investigate the performance of devices with ‘looped’ IV lines, as is sometimes recommended for clinical use and may affect pull-out resistance performance.”

Lines 109, 120, 133: The text now specifies that only participant’s right arms were used in phase 1 and in the angled testing of phase 2, and that the devices were randomly assigned to each arm in the perspirant testing in phase 2. The right arm was used during angled testing for laboratory space constraints, but this constraint was not present for pull-out testing, so the devices were randomly assigned to arms for perspirant pull-out so that only one round of perspiration was required per participant. The following text has also been added to the ‘study limitations’ section of the discussion:

“Furthermore, differences in device performance when applied to the left or right arm, or to the dominant or non-dominant arm, were not measured in this study. Arm dominance may affect the performance of these devices due to differences in size, perspiration, and hair density. Future studies should investigate the effect of arm dominance on device performance, as devices are most commonly applied to the non-dominant arm in clinical settings.”

Line 101: The outer diameter of the infusion line has now been specified in the text.

Line 113: The following text was added to clarify why phase 1 testing results were used to reduce testing time and the time-commitment of participants, as was recommended by the internal ethical review panel:

“Device performance in Stage 1 was used to select devices to be tested in Stage 2 to reduce experiment time.”

Results:

The exact wording of the comfort survey has been added to the methodology section and is copied below:

“Please provide a ranking for each of the tested devices based on your perception of their comfort during the testing, with 1 being the most comfortable and 4 being the least comfortable.”

Discussion:

Additional considerations of study limitations and future studies have been added to the discussion section.

Conclusion:

The following text has been added to the conclusion section:

“These findings should be investigated further with a larger sample size, and further parameters should be investigated, such as the effect of the line securement arrangement used and the influence of hand-dominance on device stability.”

References:

The referencing has been changed to the journal’s style.

Response to reviewer 2:

We thank the reviewer for their comments. We hope that the changes in the manuscript and the responses below sufficiently address the reviewer’s concerns.

With regards to the alignment of the problem statement and study objectives, the introduction section has been modified to clarify that the objective is the securement of IV lines, in order to reduce PVAD failure rates.

With regards to the small sample size used, the following text has been added to the conclusion section, in addition to further discussion on general study limitations throughout the discussion section:

“These findings should be investigated further with a larger sample size, and further parameters should be investigated, such as the effect of the line securement arrangement used and the influence of hand-dominance on device stability.”

With regards to the statistical significance of the comparisons presented in Figure 6, the Figure has been modified to improve the clarity of the statistical significances. Statistically significant paired comparisons are denoted with a ‘*’, and non-significant paired comparisons are denoted with ‘ns’, as described in the figure caption.

With regards to the study’s methodology and data interpretation, please see the revised manuscript where further details and clarifications have been added where necessary. The study’s methodology has been reported fully and accurately, and all data collected in this study are made available at Zenodo for reviewers and editors, and will be made openly accessible upon manuscript acceptance.

We agree with the reviewer that the problem of PVAD failure described in the introduction is a greater problem than has been addressed in our study. Our study is a step towards addressing this problem, beginning with the mechanical aspect of line securement and the best methodologies for comparing mechanical performance. We indeed believe that a larger scale trial comparing these devices in a clinical setting would be best-equipped to address the overall problem, and we believe that our study can be a step towards this direction in that it can narrow-down possible devices, as well as inform future regulatory pre-clinical testing. The following text has been added to the discussion to highlight these points:

“This study addresses the mechanical performance and testing of securement devices, but other mechanisms of PVAD failure, such as phlebitis and occlusion, remain unaddressed. A large-scale clinical trial comparing all-cause PVAD failure rates when using different securement devices should be performed. Data from sub-populations, such as those likely to perspire during treatment, can additionally address specific securement device recommendations for these sub-populations. This study is a step towards addressing the overall problem of PVAD failure and can be used to inform device selection in a future clinical trial, as well as to inform methodologies of future mechanical studies investigating PVAD dislodgement, and regulatory mechanical testing protocols.”.

---

## [Decision Letter · Decision Letter 1]

11 Jan 2026

Evaluation of peripheral intravenous line securement devices under clinically relevant loading and perspiratory conditions

PONE-D-25-39960R1

Dear Dr. Masouros,

We’re pleased to inform you that your manuscript has been judged scientifically suitable for publication and will be formally accepted for publication once it meets all outstanding technical requirements.

Kind regards,

Federica Canzan

Academic Editor

PLOS One

Additional Editor Comments (optional):

Reviewers' comments:

Reviewer's Responses to Questions

**Comments to the Author**

Reviewer #1: (No Response)

2. Is the manuscript technically sound, and do the data support the conclusions?

Reviewer #1: Yes

3. Has the statistical analysis been performed appropriately and rigorously?

Reviewer #1: Yes

4. Have the authors made all data underlying the findings in their manuscript fully available?

Reviewer #1: Yes

5. Is the manuscript presented in an intelligible fashion and written in standard English?

Reviewer #1: Yes

Reviewer #1: General comment:

Thank you for submitting your revised manuscript and responding to my previous reviewer comments. There are a few minor specific items, shown below that require consideration before publication.

Specific comments:

I note your response to my previous review suggestion stating that you have changed the title to use ‘infusion line’. However, this is not evident in the revised clean and track changes copy of the manuscript. The word ‘infusion’ is missing, consider revising the title to use the phrase ‘intravenous infusion line securement’ as to my mind this adds further clarity for readers.

Line 60 track changes (line 54 clean copy), I wonder if a minor revision to the sentence beginning: ‘Accordingly ….’ might better set the scene for readers and delineate your use of the word ‘securement’ to refer to infusion line securement in the rest of the manuscript. Perhaps: ‘Accordingly adjunct IV infusion line securement devices and techniques are widely used to anchor IV lines to the patient to reduce the risk of PVAD failure’.

Line 272 track changes (line 260 clean copy), perhaps additional sub-populations that present additional challenges for real world care complexities when securing IV infusion lines could be identified here – think about ambulant patients, confused/noncompliant patients, and children.

**Do you want your identity to be public for this peer review?** For information about this choice, including consent withdrawal, please see our Privacy Policy

Reviewer #1: No

---

## [Editor Report · Acceptance letter]

PONE-D-25-39960R1

PLOS One

Dear Dr. Masouros,

I'm pleased to inform you that your manuscript has been deemed suitable for publication in PLOS One. Congratulations! Your manuscript is now being handed over to our production team.

Kind regards,

on behalf of

Professor Federica Canzan

Academic Editor

PLOS One